# An Integrated Multi-Omics Analysis Identifying Immune Subtypes of Pancreatic Cancer

**DOI:** 10.3390/ijms25010142

**Published:** 2023-12-21

**Authors:** Yongcheng Su, Fen Wang, Ziyu Lei, Jiangquan Li, Miaomiao Ma, Ying Yan, Wenqing Zhang, Xiaolei Chen, Beibei Xu, Tianhui Hu

**Affiliations:** 1Xiamen Key Laboratory for Tumor Metastasis, Cancer Research Center, School of Medicine, Xiamen University, Xiamen 361102, China; 24520210157060@stu.xmu.edu.cn (Y.S.); 24520190154793@stu.xmu.edu.cn (F.W.);; 2CAS Key Laboratory of Quantitative Engineering Biology, Shenzhen Institute of Synthetic Biology, Shenzhen Institute of Advanced Technology, Chinese Academy of Sciences, Shenzhen 518055, China; 3Shenzhen Research Institute of Xiamen University, Shenzhen 518057, China

**Keywords:** tumor microenvironment, immunotherapy, pancreatic cancer, bioinformatic analysis, multi-omics analysis

## Abstract

Limited studies have explored novel pancreatic cancer (PC) subtypes or prognostic biomarkers based on the altered activity of relevant signaling pathway gene sets. Here, we employed non-negative matrix factorization (NMF) to identify three immune subtypes of PC based on C7 immunologic signature gene set activity in PC and normal samples. Cluster 1, the immune-inflamed subtype, showed a higher response rate to immune checkpoint blockade (ICB) and had the lowest tumor immune dysfunction and exclusion (TIDE) scores. Cluster 2, the immune-excluded subtype, exhibited strong associations with stromal activation, characterized by elevated expression levels of transforming growth factor (TGF)-β, cell adhesion, extracellular matrix remodeling, and epithelial-to-mesenchymal transition (EMT) related genes. Cluster 3, the immune-desert subtype, displayed limited immune activity. For prognostic prediction, we developed an immune-related prognostic risk model (IRPM) based on four immune-related prognostic genes in pancreatic cancer, RHOF, CEP250, TSC1, and KIF20B. The IRPM demonstrated excellent prognostic efficacy and successful validation in an external cohort. Notably, the key gene in the prognostic model, RHOF, exerted significant influence on the proliferation, migration, and invasion of pancreatic cancer cells through in vitro experiments. Furthermore, we conducted a comprehensive analysis of somatic mutational landscapes and immune landscapes in PC patients with different IRPM risk scores. Our findings accurately stratified patients based on their immune microenvironment and predicted immunotherapy responses, offering valuable insights for clinicians in developing more targeted clinical strategies.

## 1. Introduction

Among all cancers, pancreatic cancer (PC) is the most devastating, and its incidence has been steadily increasing in recent years [1]. It is predicted to become the second leading cause of cancer mortality by 2030 [2]. PC progresses rapidly, and patients often do not exhibit specific early symptoms, resulting in the cancer being frequently diagnosed at an advanced stage. Unfortunately, only about 11% of patients survive more than five years [3]. This poses significant challenges for PC research and treatment. Currently, surgical resection and chemotherapy are the main treatment options for pancreatic cancer. However, approximately 80–85% of patients with PC are not eligible for surgery at the time of diagnosis [4]. Conventional cytotoxic chemotherapy is the standard treatment, but nearly all patients with PC develop resistance, resulting in only a few months of survival benefit [5]. Furthermore, due to the high heterogeneity of PC and individual differences, the existing staging systems, subtype schemes, and prognostic markers often fail to effectively guide treatment decisions [6,7,8]. Therefore, there is an urgent need to identify novel subtypes or prognostic biomarkers to develop more effective therapeutic strategies and provide better clinical guidance.

Previous studies have shown that the tumor microenvironment (TME) not only affects tumor development, but also plays a crucial role in drug resistance in pancreatic cancer patients. This includes the regulation of cytotoxic chemotherapy and immunoregulatory therapy [9,10,11]. The TME in pancreatic cancer consists of cancer-associated fibroblasts (CAFs), immune cells, and extracellular matrix (ECM). Monocytes and neutrophils derived from the bone marrow in the TME have a significant impact on tumor proliferation, invasion, and response to immunotherapy and chemotherapy. For instance, certain myeloid-derived suppressor cells secrete TGF-β, COX-2, and IL-10, leading to the immunosuppression of T cells [12,13]. Additionally, cancer-related fibroblasts secrete immune-related inflammatory factors like IL-6, CXCL12, and TGF-β, which influence the phenotype of pancreatic cancer, immune infiltration of T cells, and the efficacy of current treatment regimens [14,15,16]. Although immunotherapy has shown promising results in various tumors [17,18,19,20], its effectiveness in pancreatic cancer has not met expectations [21]. Pancreatic cancer is characterized by a highly immunosuppressive and interstitial fibrotic TME with limited infiltration of effector T cells. Some researchers have shown that changing the degree of T cell infiltration in TME could be the key to improving the efficacy of immunotherapy in pancreatic cancer [6,22,23]. Therefore, conducting a comprehensive analysis of the immune TME in pancreatic cancer to identify new subtypes and their unique characteristics could help assess pancreatic cancer risk, aid prognosis, and guide medication. Furthermore, it may also aid in the identification of novel therapeutic targets, potentially leading to improved outcomes for some patients.

In this study, we identified three immune PC subtypes by analyzing the activity variation of C7 immunologic signature gene sets in PC and normal samples: immune-desert, immune-excluded, and immune-inflamed subtypes. The overall survival and clinicopathological features of PC patients with different subtypes were significantly different. To more accurately evaluate the prognosis and treatment of a single patient, we established the immune-related prognostic risk model (IRPM) score by further screening the immune prognostic gene sets from these three subtypes, including RHOF, CEP250, TSC1, and KIF20B. The score effectively predicted the prognosis of PC patients, and external validation results showed that IRPM could be used to determine immunotherapy regimens for PC. Subsequently, we confirmed that knocking down RHOF reduced the proliferation, migration, and invasion of PC cells in vitro. RHOF, a member of the Rho GT3 family, was a risk factor in the scoring formula and has been involved in regulating various cell functions, such as cell adhesion and remodeling of pseudopods [24]. Launce G et al. showed that B-cell-derived tumor cells and tissues expressed higher levels of RHOF than benign cells [25]. Li et al. found that RHOF is upregulated in hepatocellular carcinoma and plays a key role in promoting cancer cell migration, invasion, and EMT [26]. We observed significant effects of RHOF on the proliferation, migration, and invasion of PC cells in vitro. Additionally, we comprehensively analyzed the somatic mutation and immune landscapes in PC patients with different IRPM risk scores, which have significant implications for clinicians in developing more targeted strategies.

## 2. Results

### 2.1. Identification of Immune Subtypes of PAAD Patients

A flow chart of the study design is presented in Figure 1. We first collected the TCGA-PAAD dataset and C7 immunologic and hallmark gene sets from TCGA and GSEA databases, respectively. Then, GSVA analysis was used to evaluate the variation of the immunologic signature gene sets (C7) in PAAD patients. Appendix A shows enrichment scores in PAAD and adjacent normal tissues.

Based on the enrichment scores, PAAD patients were divided into distinct clusters. The “factoegxtra” package was subsequently used to determine the optimal number of clusters (Figure 2A,B). Finally, the PAAD patients were grouped into three clusters using the non-negative matrix factorization (NMF) method; survival analysis revealed that cluster 3 had better prognosis than clusters 2 or 1 (*p* < 0.05, Figure 2C). Figure 2D shows the clustering heatmap for three subtypes, and Figure 2E displays the silhouette width of the NMF clustering, with an average silhouette width of 0.61, indicating that our clustering was stable. The external validation dataset (GSE62452) was used to validate the clustering algorithms. Interestingly, PAAD patients from the GSE62452 dataset were also divided into three clusters, and significant differences were found in overall survival among the three clusters, which showed that our clustering was stable and reliable (Appendix A).

### 2.2. Immune Landscape in PAAD Patients among Three Clusters

Figure 3A and Appendix A illustrate the significant differences observed in grade and stage among the cluster groups. Subsequently, we analyzed immune cell infiltration levels among the three clusters by the ssGSEA algorithm (Figure 3B), and significant distinctions were observed among the three clusters. Cluster 2 showed a higher immune cell infiltration, especially for activated B cells, CD8+ T cells, DCs, NK cells, and monocyte. The estimated algorithm was further applied to evaluate the TME score in patients with PAAD. The results also showed that cluster 2, with high estimate (Figure 3C) and immune scores (Figure 3D) and high infiltration levels of immune cells, was correlated with worse prognosis. To further explore roles of the three clusters in immunomodulation, we analyzed the cytokine and chemokine expression of the three clusters: immune activation (IA)-related genes (GZMA, IFNG, TNF, CXCL9, PRF1, GZMB, TBX2, CD8A, and CXCL10); immune checkpoint (IC)-related genes (PDCD1, LAG3, TNFRSF9, CD86, HAVCR2, CD80, IDO1, TIGIT, CD274, and CTLA4); and TGF)-β/EMT signaling pathway-related genes, including VIM, COL4A1, CLDN3, ACTA2, TGFBR2, ZEB1, and TWIST1. All related cytokines and chemokines were derived from published studies [27,28,29]. As displayed in Figure 3E,F, the expression of IA and IC-related genes in cluster 2 was elevated, indicating that the TME exerts a key role in cluster 2. Previous studies showed that TGF-β/EMT was directly linked to poor prognosis [30,31]; furthermore, increased levels of TGF-β/EMT signaling pathway-related genes (Figure 3G) and stromal scores (Figure 3H) suggest higher degrees of stromal cell infiltration in cluster 2 PAAD. Moreover, TIDE scores were markedly elevated in cluster 2 (Figure 3I). Thus, we conclude that cluster 2 PAAD may stem from tumor-infiltrating immune and stromal cells instead of cancer cells.

### 2.3. Correlation Analysis between Clinicopathologic Features among Clusters

To identify the association between clinicopathological features among the three clusters, KEGG enrichment analysis was conducted. Compared to clusters 1 and 3, cluster 2 was obviously enriched in stromal and cancer-related pathways, comprising TGF-β, MARK, cell adhesion, Notch, and JAK/STAT signaling pathways (Figure 4A–C, Appendix A). Furthermore, to explore the key gene sets of each cluster, differential scores of gene sets were calculated among the three clusters and intersected. Finally, we obtained nine differential gene sets as follows: ova alone vs. ova with mpl immunized mouse whole spleen 6 h up, ra vs. untreated tconv up, hallmark mitotic spindle, resting vs. no treated cd4 tcell dn, hallmark notch signaling, immature vs. listeria inf mature dc dn, hallmark g2m checkpoint, classical m1 vs. alternative m2 macrophage dn, and naive bcell vs. bm plasma cell dn; the heatmap demonstrated clinicopathologic features among the nine differential gene sets (Figure 4D).

To further identify prognosis-related gene sets in patients with PAAD, univariate Cox analysis for the prognostic value of nine differential gene sets was carried out. The three gene sets were closely related to the overall survival (OS) of PAAD patients: hallmark g2m checkpoint, ova alone vs. ova with mpl immunized mouse whole spleen 6 h up, and hallmark mitotic spindle. The gene sets, hallmark g2m checkpoint, and thallmark mitotic spindle were further screened by using LASSO regression analysis (Figure 5A,B), and the survival curve also revealed that the risk gene sets with high enrichment scores correlated with poorer prognosis (Figure 5C,D). Subsequently, 360 hub genes from prognosis-related gene sets were determined.

### 2.4. GO and KEGG Pathway Enrichment Analysis 

To explore how prognosis-related genes exert their function, GO and KEGG pathway enrichment analyses were subsequently conducted (Appendix A). Figure 5E shows the 10 highest-ranking GO in terms of molecular function (MF), biological process (BP), and cellular component (CC). The prognosis-related genes were mostly related to tubulin, microtubule, actin, guanyl-nucleotide exchange factor, GTPase activator, rho GTPase, ras GTPase, small GTPase, rho guanyl-nucleotide exchange factor, and ras guanyl-nucleotide exchange factor activities. KEGG enrichment analysis indicated that these genes were related to cancer-related pathways, such as pancreatic cancer, TGF-β, focal adhesion, and Ras signaling (Figure 5F). 

### 2.5. Construction of a Prognostic Model for PAAD Patients

Given PC’s high heterogeneity and complexity, our immune subtypes could not accurately evaluate risk stratification in individual samples; thus, we built a scoring system to comprehensively assess the risk scores of each PC patient—immune-related prognostic risk model (IRPM). We identified 105 immune-related genes from 360 hub genes as immune-related prognostic genes (IRPGs) in PAAD patients using univariate Cox analysis (Appendix A). Further, LASSO regression analysis was performed to screen IRPGs from the Cox analysis results, resulting in the extraction of 19 prognostic genes (Figure 6A,B). Surprisingly, RHOF exhibited the highest importance among the 19 prognostic genes, as determined by the random forest algorithm (RF) (Figure 6C). Additionally, six genes (RHOF, KIF20B, PML, ARHGAP29, FLNB, and CEP250) were further screened using the RF algorithm (Figure 6D). Considering that fewer variables could make the proposed model easier to control, ultimately, four genes, namely, RHOF, CEP250, TSC1, and KIF20B, were included in the immune-related prognostic risk model (IRPM). The IRPM risk scores were calculated as follows: risk score = 1.31 × KIF20B + 0.79 × RHOF − 1.30 × CEP250 − 0.79 × TSC1. Based on the median risk score, PAAD samples were separated into high-risk and low-risk groups. The IRPM high-risk group had a higher incidence of death and worse prognosis than the IRPM low-risk group (Figure 6E). The external validation dataset (GSE62452) was used to test the external validity of the IRPM, and the results also showed that our risk model, IRPM, was stable in both the internal and external cohorts (Figure 6F,G). Moreover, significant differences were observed in the T stage between the IRPM high-risk and low-risk groups. The IRPM score increased as the T stage advanced (Figure 6H).

### 2.6. Prediction Performance and Independence Evaluation of IRPM Model 

We then performed ROC analyses to evaluate the performance of our IRPM model. The area under the ROC curve (AUC) of the 1-year, 2-year, and 3-year ROC curves was 0.757, 0.766, and 0.763, respectively (Figure 7A). The ROC curve also indicated that our IRPM had a good capability to predict the prognosis of PAAD patients compared to clinicopathological features, including age, sex, grade, clinical stage, and TNM stage (Figure 7B). Univariate Cox regression (Figure 7C) and multivariate Cox regression also demonstrated that IRPM was an independent prognostic factor in patients with PAAD (Figure 7D). Furthermore, we compared our model with other previously developed models [32,33,34,35], and significant improvement in the estimation of survival was achieved with the continuous form of IRPM relative to other previously developed models, and the c-index indicated that our IRPM had a higher predictive value (C-index: 0.707 vs. 0.601, 0.707 vs. 0.641, 0.707 vs. 0.596, and 0.707 vs. 0.703, respectively) (Figure 7E,F).

Next, we used TCGA pan-cancer gene expression data to evaluate the performance of the IRPM models in predicting cancer types. Results of the Cox regression analysis showed the IRPM scores to be various survival indicators, including DSS, overall survival (OS), and progression-free interval (PFI), in most of independent TCGA cancers (Figure 7G).

The IRPM score demonstrates a high accuracy in predicting survival and can be utilized to assess the prognosis of patients diagnosed with pancreatic cancer. This highlights the advantages of using the IRPM score for prognostic risk assessment in pancreatic cancer patients.

### 2.7. In Vitro Experiments Validated That RHOF Has Tumor-Promoting Effects 

In light of the accuracy of the IRPM score in the prognostic assessment of PC patients and the significance of the RHOF gene in prognostic models, we conducted a series of in vitro experiments to investigate its role in the occurrence and progression of pancreatic cancer. The lentiviral packaging of shRHOF was employed to infect pancreatic cancer cell lines, resulting in a successful decrease in RHOF expression at both the RNA and protein levels (Figure 8A,B). Subsequently, the growth rate and clone formation ability of MIA and PANC-1 cells were significantly reduced upon decreased RHOF expression, as evidenced by MTT and plate clone formation assays (Figure 8C,D). Moreover, the utilization of the EdU assay confirmed a noticeable deceleration in the growth of pancreatic cancer cells due to decreased RHOF expression (Figure 8E,F). To investigate the influence of RHOF on invasion and metastasis in pancreatic cancer cells, alterations in the invasion and migration capabilities of these cells were examined in stable cell lines with RHOF knockdown. Scratch assays revealed a substantial inhibition of cell migration in MIA and PANC-1 cells upon the knockdown of RHOF (Figure 8G,H), which was further validated by transwell assays, indicating a significant decrease in the invasion and migration capacities of these cells (Figure 8I,J). 

These findings highlight the regulatory function of RHOF, possibly as an oncogene, in promoting the proliferation, invasion, and migration abilities of pancreatic cancer cells. RHOF, being the most significant factor in the IRPM score, may contribute to the progression of pancreatic cancer. This also confirms the reliability of the IRPM score in assessing the risk of pancreatic cancer to a certain extent.

### 2.8. Association of RHOF Expression with the Tumor Immune Microenvironment 

To further clarify the possible role of RHOF in the tumor microenvironment, we analyzed single-cell sequencing data through the TISCH database (a scRNA-seq database that provides extensive cell type annotations at the single-cell level, allowing for TME exploration across various cancers) [36]. We collected nine public datasets (PAAD_ CRA001160, PAAD_GSE111672, PAAD_GSE141017, PAAD_GSE148673, PAAD_GSE154763, PAAD_GSE154778, PAAD_GSE158356, PAAD_GSE162708, PAAD_GSE165399) based on the TISCH database. As shown in Appendix A, we obtained a consistent distribution of RHOF levels across cell types in different datasets. The most prominent senescence levels were exhibited in immune cells (CD8+T cells). Subsequently, we further explored and illustrated the mechanism of the RHOF gene in immune regulation in GSE158356 datasets. The results of UMAP showed that all the cells were labeled into six different cell subpopulations, including Acinar, B cell, CD8+T cell, epithelial, fibroblasts, and mono/macro (Appendix A). For the GSE158356 dataset, RHOF is mainly expressed in CD8+T cell (Appendix A). Previous studies have shown that CD8+T cells are major drivers of anti-tumor immunity [37,38], which suggests that RHOF expression is related to pancreatic cancer and the immune microenvironment. And functional enrichment analysis subsequently conducted further confirmed our speculation that the activity of the hallmark gene sets MYC targets V1 pathways in RHOF high-expressing cell cluster (CD8+T cell) was significantly increased (Appendix A). 

We also performed GSEA analysis using TCGAPAAD bulk RNA-seq data to compare the expression level of RHOF concerning related signaling pathways. The cancer-associated pathway signatures were extracted from Hu et al. [39], the cancer-immunity cycle reflects the anticancer immune response [40], and the activation levels of the cancer-immunity cycle were retrieved from tracking tumor immunophenotype (TIP) (http://biocc.hrbmu.edu.cn/TIP/, accessed on 14 February 2022) [41].

And as shown, RHOF was significantly positively correlated with oncogenic pathways (such as APM_signal, cell cycle, DNA replication, p53 signaling pathway) (Appendix A). Interestingly, we further found that RHOF is positively correlated with cancer immunity cycle pathways, like CD8+T cell, TH22, TH1, and MDSC; recruiting; infiltration of immune cells into tumors; and recognition of cancer cells by T cells, which further confirmed that RHOF is closely related to the immune microenvironment (Appendix A). Moreover, subsequent IPS analysis also showed higher IPS scores in the RHOF high-expression group (Appendix A), which meant that the RHOF high-expression group was more sensitive to immunotherapy, especially anti-CTLA-4 and anti-PD-1 treatments (Appendix A). All the above results indicate that RHOF participates in tumor invasion and metastasis and plays an important role in immune regulation, which is consistent with our previous results.

### 2.9. Immune Landscape in PAAD Patients with Different IRPM Risk Scores

To explore the correlation between immune cell infiltration and IRPM risk scores, we first conducted Spearman’s correlation analysis based on the EPIC, QUANTISEQ, XCELL, CIBERSORT-ABS, MCPcounter, TIMER, and CIBERSORT algorithms. The results of the correlation analysis were visualized using a lollipop plot, which showed that the IRPM risk score was inversely correlated with the infiltration of multiple anti-cancer active ingredients, including CD4+T cells, B cells, NK cells, CD8+T cells, and DCs (Figure 9A). We also found that the infiltration level of CD8+T cells, B cells, and NK cells decreased in the IRPM high-risk group (Appendix A). Notably, we discovered that the stromal and microenvironment scores were inversely correlated with IRPM scores (Appendix A). Moreover, using the ESTIMATE algorithm, we also obtained the same results that the IRPM risk score inversely correlated with immune, estimate, and stromal scores (Figure 9B–D, *p* < 0.01), indicating that a higher risk of IRPM is associated with less intratumorally infiltrated stromal cells. Differential analysis was subsequently conducted to investigate the association between IRPM and immune checkpoints. PD-1 and CTLA-4 expressions were found to be higher in the IRPM low-risk group than in the IRPM high-risk group (Figure 9E,F), whereas no significant difference was observed for PD-L1 (Figure 9G).

As previously mentioned, the TIDE algorithm [42], the immunophenoscore (IPS) score [43], and subclass mapping [44] were used to predict responses to the immune checkpoint blockade (ICB). TIDE scores were markedly lower in the IRPM high-risk group, which indicated a low immune escape and better response to immunotherapy (Figure 9H). A subsequent IPS analysis also showed higher IPS scores in the IRPM high-risk group (Figure 9I), which also meant that the IRPM high-risk group was more sensitive to immunotherapy, especially CTLA-4 treatment (Appendix A), although there was no statistical difference (*p* > 0.05). In addition to the TIDE and IPS predictions, we also used subclass mapping to compare the expression profile of the two IRPM score groups; we performed the definitions with another published dataset containing 47 patients with melanoma that responded to immunotherapies. We were very delighted to see that the IRPM high-risk group was more promising to respond to anti–CTLA-4 therapy (nominal *p* = 0.017) (Figure 9J). Taken together, these results suggest that IRPM may be a potential marker of the immunotherapy effect of PAAD patients.

### 2.10. Single-Cell Analyses Reveal IRPM Score Heterogeneity and Associated Immune Characteristics in Tumor Microenvironments

To test for heterogeneity in IRPM score levels in TME, we downloaded the pancreatic cancer single-cell sequencing dataset GSE212966 from the GEO database, extracted data from six patients with primary pancreatic cancer in the dataset, and analyzed them using the R package “Seurat”. Low-quality cells were identified and filtered using the following criteria: (1) nFeatures < 200 and nFeature > 2500; (2) mitochondrial gene ratio > 10%. A total of 30629 high-quality cells were used for analysis, and we identified nine cell clusters: B_cell, Endothelial cells, Neutrophils, T_cells, Chondrocytes, Monocyte, NK_cell, Tissue_stem_cells and Epithelial_cells (Figure 10A). Subsequently, we assessed the differences in IRPM scores across cell types, and, interestingly, we found that for IRPM scores versus normal samples, tumor patients exhibited more significant levels of IRPM scores (Figure 10B). Similarly, we observed intercellular heterogeneity in IRPM scores in the microenvironment of pancreatic cancer patients, particularly in Neutrophils, T_cells, Chondrocytes, NK_cells, Tissue_stem_cells, and Epithelial_cells (Figure 10C), and the above results further indicate the impact of intratumor IRPM heterogeneity in the tumor microenvironment.

We subsequently focused on intercellular interactions. Based on their median IRPM scores, epithelial cells were divided into high-IRPM and low-IRPM groups. We found that epithelial cells in the low-IRPM group preferentially communicated with different neighboring cells (Figure 10D). A total of 27 significant pathways were detected, among which the TWEAK signaling pathway had a higher relative strength index in incoming signaling patterns, playing a major roles as a receiver and influencer in the IRPM-low Epithelial_cells cluster (Appendix A). Furthermore, epithelial cells in the low-IRPM group had a high number of interacting pairs with Monocyte, Tissue_stem_cells, endothelial cells, NK_cell, and T_cells, especially for SPP1_CD44, ITGAV_ITGB5, ITGAV_ITGB1, ITGA4_ITGB1, ITGA5_ITGB1, ITGA6_ITGB1, and NAMPT_INSR (Figure 10E), which are known to promote tumor growth, invasion, and metastasis. 

The pathway enrichment analysis of DEGs in high- and low-IRPM score groups indicated the KEGG pathway enrichment of DEGs in epithelial cells between the high-IRPM and low-IRPM groups, also revealing the significant enrichment of cancer-related pathways in epithelial cells, including adherens junction, pathways in cancer, tight junction, and leukocyte transendothelial migration (Figure 10F). Moreover, immune-related pathways were significantly enriched. Enrichments included the following: untreated vs. tgfb il6 treated cd4 tcell up, resting vs. tcr activated cd4 tcell up, bm vs. colon tumor myeloid-derived suppressor cell dn, flt3l induced dec205 pos dc vs. cd4 tcell dn, unstim vs. anti igm stim tak1 ko bcell 3h dn, wt vs. klf13 ko thymic-memory-like cd8 tcell dn, and marginal zone bcell vs. memory bcell day7 dn, especially the low-IRPM group (Figure 10F), which suggests that IRPM score was closely associated with the TME and tumor progression in PAAD.

### 2.11. Comparison of Somatic Mutation Landscape in PAAD Patients with Different IRPM Risk Scores

The somatic mutation landscape analysis indicated that more mutational events occurred in the IRPM high-risk cohort, and KRAS was the most frequently mutated gene, followed byTP53, in PAAD samples (Figure 11A,B). Moreover, the IRPM high-risk-score group showed a more extensive tumor mutation burden (TMB). The TMB analysis confirmed that the IRPM high-risk cohort was closely related to a higher TMB (Figure 11C), which was associated with worse survival in PAAD (Figure 11D). Further survival analysis showed that the high-IRPM cohort with a high TMB status had the worst clinical outcomes among all subgroups (Figure 11E).

It was observed that a higher expression of protective prognostic factors (CEP250 and TSC1) in cluster 3 showed better clinical outcomes, while high expression levels of risk factors (RHOF and KIF20B) in cluster 1 correlated with worse prognosis (Figure 11F). The gene regulatory network was used to present the interactions of four prognostic genes, regulatory relationships, and prognostic significance for PAAD (Figure 11G). In agreement with our findings, cluster 1 was associated with a high IRPM risk score, with more deaths (Figure 11H,I).

### 2.12. Chemotherapy Drug Sensitivity Analysis in PAAD Patients with Different IRPM Risk Scores

To further compare chemotherapy drug sensitivity in patients with PAAD for different scores, we conducted drug sensitivity analysis using the GDSC database. As shown in Figure 10, the IC_50_s of AICAR (Figure 12A), Gefitinib (Figure 12B), Bleomycin (Figure 12C), and Gemcitabine (Figure 12D) were lower in PAAD patients with low-risk scores, indicating that patients in the low-score cohort seemed to be more sensitive to AICAR, Gefitinib, Bleomycin, and Gemcitabine. Therefore, these antitumor drugs should be prioritized in the IRPM low-score cohort. Patients in the high-score cohort seemed to be more sensitive to Sunitinib (Figure 12E), Imatinib (Figure 12F) and Bexarotene (Figure 12G). Lower IC_50_s were observed in the high-risk group, which suggests that increased risk was accompanied by increased sensitivity to Sunitinib, Imatinib, and Bexarotene. Thus, these drugs should be prioritized for use in high-risk patients.

## 3. Discussion

Thus far, there have been numerous studies identifying novel cancer subtypes, with these studies being mostly in the field of oncology [45,46,47,48,49,50]. Only few studies have attempted to focus on the role of adjacent normal tissues. The present study attempted to investigate novel PAAD subtypes based on the variation of immunologic signature gene sets in tumor and adjacent normal tissues, and three PAAD immune-related subtypes were identified according to the NMF algorithms. Notably, a better clinical outcome was observed in patients with subtype 3 PAAD than in those with other subtypes. Moreover, three prognostic-associated gene sets were identified based on gene-set difference scores among three gene clusters. Subsequently, KEGG analysis showed that genes from the PAAD prognosis-related gene sets were markedly enriched in cancer-related signaling pathways, comprising pancreatic cancer, TGF-β, focal adhesion, and ras signaling pathway. 

Previous studies have revealed that the TGF-β signaling pathway can induce EMT, thus upregulating PD-L1 expression and contributing to cancer invasion, metastasis, and immune escape [51,52,53,54]. In line with previous findings, increased levels of TGF-β/EMT signaling pathway-related genes were observed in cluster 2 PAAD samples, accompanied by an increase in PD-L1 levels. ICB has seen remarkable progress in the treatment of multiple malignancies [55]. To predict ICB responses, we calculated TIDE scores for TCGA-PAAD samples using an online web browser (http://tide.dfci.harvard.edu/, accessed on 14 February 2022) [42]. Our study also demonstrated that TIDE scores in the cluster 2 PAAD group were markedly increased compared to those in other groups, indicating a high immune escape and poor response to immunotherapy. In agreement with our findings, recent studies showed that TGF-β can interact with PD-L1, which inhibits T cell activation and induces and sustains the function of Tregs [56,57]. Furthermore, PD-L1 acts independently and synergistically with TGF-β to produce immune tolerance, resulting in the poor efficacy of ICB. Therefore, novel immunotherapy targeting PD-L1 and TGF-β pathways would be an effective therapeutic strategy for patients with high levels of TGF-β and PD-L1, which have achieved impressive clinical effects [56,57]. Also, to further increase the reliability of the results, we reanalyzed our dataset after eliminating the potentially contaminated samples. According to the results, even after the removal of these samples and by using solely pure pancreatic cancer samples, the conclusions derived from the new analyses remained consistent with the findings previously outlined in our article. The results of this analysis are attached as an additional file (see Appendix A).

To better predict the immunotherapeutic effect of tumors, previous studies classified most human tumors into three subtypes based on the characteristics of the TME as follows: immune-desert, immune-excluded, and immune-inflamed subtypes [28,58]. The immune-inflamed subtype had better response rates to immune checkpoint therapy; the immune-excluded subtype was marked by massive infiltration of stromal components and the presence of numerous immune cells; the immune-desert subtype was related to a lack of T cells in the tumor. Notably, our clusters were highly consistent with previous immunophenotyping, with cluster 2 showing a strong linkage to stromal activation, including high expression levels of TGF-β (VIM, COL4A1, CLDN3, ACTA2, TGFBR2, ZEB1, and TWIST1), EMT (SFRP1 and ACTA2), cell adhesion (PDGFRA and GREM1), and extracellular matrix remodeling (MYH11, FOXF2, TIMP2, and DCN) related genes, which were considered the immune-excluded subtype. Meanwhile, cluster 1, characterized by a higher response rate in ICB, had the lowest TIDE scores among the 3 clusters and was classified as the immune-inflamed subtype; cluster 3 was defined as the immune-desert subtype.

Because biological features differ among distinct PC clusters, different treatment strategies are required. Thus, for cluster 1 PAAD samples, ICB therapy should be prioritized. Cluster 2 is highly correlated with stromal activation, especially TGF-β pathways; thus, simultaneously targeting TGF-β pathways and PD-1/PD-L1 with combination treatments might achieve impressive clinical effects [56,57]. Regarding cluster 3, recent studies have pointed out that immune-desert subtype patients could benefit from a combination of low-dose radiotherapy (LDRT) and immunotherapy [58].

The clinical outcome for patients with PAAD closely correlated with multiple factors, including age, sex, grade, and stage [59]; however, no single factor has been reported as an independent prognostic factor in PAAD. In this study, 105 immune-related genes from tumor gene sets were identified as IRPGs in PAAD patients by the univariate Cox analysis, and 4 genes of IRPGs, RHOF, CEP250, TSC1, and KIF20B were then used to construct the IRPM. Notably, the IRPM demonstrated a better predictive capacity in determining PAAD prognosis compared to other clinical characteristics; this was also successfully validated by an external cohort. Moreover, compared with the prognostic models for pancreatic cancer constructed by previous investigators, our IRPM demonstrated a higher C-index, which further suggests the reliability of our IRPM score. More importantly, the crucial gene in the prognostic model, RHOF, a member of the Rho GTP enzyme family, has been extensively studied for its role in regulating tumor cell adhesion, migration, epithelial-mesenchymal transition, and drug resistance [60,61,62,63]. However, its association with the regulation of the TME remains unclear. Our research demonstrates that knocking down RHOF effectively reduces the proliferation, invasion, and metastasis of PC. Furthermore, we found a positive correlation between RHOF expression and the activity of several steps in the cancer immune cycle. The cancer immune cycle represents the complex immunoregulatory interactions in the TME that contribute to the body’s immune response against cancer [40]. Our study reveals that RHOF not only enhances the tumor antigen presentation process but also leads to a significant increase in the levels of various effector tumor-infiltrating immune cells, such as CD8+T cells, TH22 cells, and TH1 cells, in the high-RHOF group. Moreover, single-cell analyses demonstrated an increased expression of RHOF in CD8+T cells, indicating its close association with the TME. We speculate that a high RHOF expression promotes antigen presentation by facilitating the release of tumor antigens and activation of T cells, thereby enhancing antitumor immune responses. Subsequent IPS analysis further confirms that PC patients with a high RHOF expression exhibit increased sensitivity to anti-PD-1/CTLA-4 ICB therapy. These findings highlight the potential of RHOF as a valuable clinical predictor for immunotherapy response in pancreatic cancer patients.

The efficacy of immunotherapy in PAAD is suboptimal partly because of the immunosuppressive TME [64,65,66]. Previous studies have revealed that CD8+T cells [67,68,69,70], NK cells [71], and B cells, which are antitumor immune cells, are associated with better clinical outcomes. Herein, we comprehensively analyzed the correlation between TME and different IRPM risk score groups. Low infiltration levels of CD8+T cells, B cells, and NK cells were observed in the IRPM high-risk group, which correlated with worse prognosis. Concordantly, we also found that the IRPM risk scores are inversely correlated with immune, estimate, and stromal scores, and the PD-1 and CTLA-4 immune checkpoint expression was also found to be lower in the IRPM high-risk group. We conclude that the high-risk group appears to have more immune “cool” tumors with less immune cell infiltration. Moreover, the TIDE score and the IPS were also compared between high- and low-risk groups. Notably, both IPS and TIDE algorithms showed that, relatively, the IRPM high-risk group was more sensitive to immunotherapy, especially CTLA-4 treatment; thus, we suggest that CTLA-4 inhibitors, such as ipilimumab and tremelimumab, may perform antitumor functions in these patients.

Acquired resistance to currently available anticancer drugs is a principal cause of treatment failure in PAAD patients [72,73]; thus, there is an urgent need to discover novel drug combinations or therapeutic regimens. We also examined drug sensitivity to commonly prescribed chemotherapy drugs for distinct IRPM risk score groups and found that our model would provide a more targeted strategy for guiding clinicians to determine potential target treatment drugs for PAAD patients with different IRPM risk scores.

## 4. Methods and Materials

### 4.1. Data Sources

Complete clinical and expression data were retrieved from TCGA (http://portal.gdc.cancer.gov, accessed on 2 November 2021) and GEO databases (GSE62452 and GSE212966). Mutation data were retrieved from TCGA database. Immunologic and hallmark genes (c7: immunologic signature gene sets) and KEGG gene sets (c2.cp.kegg.v7.2.symbols.gmt) were obtained from GSEA (http://www.gsea-msigdb.org/gsea/downloads.jsp, accessed on 1 October 2021). 

### 4.2. Cell Lines, Cell Culture, and Cell Transfection

PANC-1, MIAPaCa-2, and HEK293T were acquired from Chinese Academy of Sciences Cell Bank of Type Culture Collection (Shanghai, China). And all cell lines were identified. The cells were cultured in Dulbecco’s Modified Eagle Medium at 37 °C in a 5% CO_2_ incubator. All media were supplemented with 10% fetal bovine serum (HyClone, Logan, UT, USA), 100 U/mL and 100 μg/mL (Life Technologies, Carlsbad, CA, USA). Lentiviral expression system was utilized to induce short hairpin RNA (shRNA) targeting RHOF. The approach for generating stable cell lines was adopted from a previous study [26]. In brief, the lentiviral vector (plko.1-shRHOF/shCtrl), PAX2, and pMD2.G plasmids were transfected into HEK293T using polyethylenimine (Invitrogen, Thermo Fisher Scientific, Shanghai, China). The cell supernatant was collected after 48 h and centrifuged to obtain the viral supernatant. PANC-1 and MIAOaca-2 cells were transduced with lentiviral supernatant within 2 μg/μL polybrene. And cells were selected by 1 μg/mL puromycin (Invitrogen, Thermo Fisher Scientific, Shanghai, China). The knock-down effect of protein was demonstrated using Western blot. The shRNA sequences were the following: CAACAAGATGAAGAGCACCAA (shCtrl), CCTGGAATGTTCCGCCAAGTT (shRHOF-1), and CAACGTCCTCATCAAGTGGTT (shRHOF-2).

### 4.3. Cell Proliferation Analysis

Cell proliferation was measured using the methyl thiazolyl tetrazoliym (MTT) assays. The MTT assays were performed following the manufacturer’s instructions. Approximately, cell lines stably transfected with shRNA were planted in 96-well plates (5 plates) at 2500 cells per well. After the cells were attached, the medium was removed and replaced with a fresh 100 μL medium containing 20 μL MMT and incubated for 4 h until the purple precipitate was fully yielded. Absorbance at 490 nm was measured 15 min after 150 μL DMSO was added into each well to dissolve the precipitate. Above steps were repeated 24, 48, 72, and 96 h later. The OD values measured in each group were compared with the average OD values measured in the first 96-well plate.

Cell proliferation was also evaluated using colony formation assays and 5-ethynyl-20-deoxyuridine (EdU) assays, following established procedures [74]. For the colony formation assays, the cells were planted in 6-well plates at a density of 500 cells per well and cultured for a duration of 2 weeks. Subsequently, After removing the cell media from the 6-well plate, cells were washed once with PBS and fixed with 4% paraformaldehyde for 15 min. They were washed with PBS 3 times for 3 min each time. Crystal violet (0.5%) staining was performed for 30 min, and the liquid was removed. It was washed with PBS 3 times for 3 min each time, followed by the quantification of colony numbers. The EdU assays involved seeding PANC-1 and MIAPaCa-2 cells at a density of 8 × 10^3^ cells/well in 96-well plates. To conduct the EdU assays, a Cell-LightTM EdU Apollo 567 In Vitro Kit (RiboBio, Guangzhou, China) was employed, adhering to the manufacturer’s instructions. The fluorescence images of cy3-EdU and Hochest were captured by confocal microscopy at 40× (Olympus IX73, Olympus, Tokyo, Japan).

### 4.4. Wound-Healing

Wound-healing experiments were conducted following established protocols [75]. Specifically, two sets of horizontal and vertical parallel lines were marked on the back of the 6-hole plate for positioning when taking photos. Then, shCtrl and shRHOF cells were separately seeded in these 6-well plates (approximately 5 × 10^5^ cells per well). After 12 h of incubation in FBS-free medium, a 10 μL pipette tip was utilized to create a scratch wound on the cell layer. The cells were gently washed with PBS and replaced with a new medium containing 0.5%FBS (these two cell lines could not be cultured without FBS for a long time). Subsequently, images of the gap closure were captured at specified time points (0 h, 24 h, and 48 h) using a microscope with a 4× magnification. The wound closure rate was quantified through the utilization of Image J software (version 1.46).

### 4.5. Migration and Invasion Assays

Transwell chambers (Corning, NY, USA) were used to perform transwell migration assays. A total of 50,000 cells were seeded in serum-free media into the upper chamber and maintained at 37 °C using a 5% CO_2_ incubator. The bottom of the chamber was filled with normal media. Matrigel (Corning, NY, USA) was used to pre-coat the chamber membrane in 24-well dishes for the invasion assays. The chambers were maintained in a 5% CO_2_ incubator at 37 °C for 24 h. Cells in the upper chamber that do not cross the membrane were gently wiped away using a cotton swab. The cells on the lower side of the filter were fixed and then stained with 0.5% crystal violet and counted under a light microscope (Olympus IX73, Olympus, Japan).

### 4.6. Western Blot Analysis

The previously described procedure was employed to conduct Western blot analysis [76]. RIPA buffer was utilized to extract proteins, followed by quantification using a protein quantification kit based on bicinchoninic acid (BCA) (Thermo Fisher Scientific, Cat. No. 23228). Protein lysates (15–30 μg) underwent separation through 10% SDS-PAGE and subsequent transfer onto a PVDF membrane (Merck Millipore Ltd., Darmstadt, Germany). Then, 10% skimmed milk was used for blocking membranes. The internal reference and target protein were detected by β-actin (1:10,000, Sigma, St Louis, MO, USA) and RHOF (1:500; Proteintech, Wuhan, China)

### 4.7. Quantitative Real-Time PCR (qPCR)

Total RNA was extracted using the Trizol reagent (Invitrogen), following the manufacturer’s guidelines. The reverse transcription system (YEASEN Biotech, Shanghai, China) was employed to synthesize cDNA from 1 μg of mRNA. Subsequently, qPCR amplification was conducted using the SYBR Master Mix (YEASEN Biotech, Shanghai, China), with the GAPDH gene serving as the reference. The primer sequences for qPCR are as follows: GAPDH-F: 5′-TGCACCACCAACTGCTTAGC-3′; GAPDH-R: 5′-GGCATGGACTGTGGTCATGAG-3′; RHOF-F:5′-CCCCATCGGTGTTCGAGAAG-3′;RHOF-R:5′-CCCCATCGGTGTTCGAGAAG-3′.

### 4.8. Gene set Variation Analysis (GSVA) 

The “GSVA” package was used to perform GSVA analysis [77]. Clustering of pancreatic adenocarcinoma (PAAD) transcriptome data was performed with the “NMF” package [78]. Gene Ontology (GO) and Kyoto Encyclopedia of Genes and Genomes (KEGG) pathway enrichment analyses were performed using the “clusterProfiler” R package [79]. The tumor immune dysfunction and exclusion (TIDE) score was evaluated using the TIDE website (http://tide.dfci.harvard.edu/, accessed on 14 February 2022).

### 4.9. Construction of Prognostic-Associated Risk Model in PC Patients

Univariable Cox regression analyses were first performed to characterize immune-related prognostic genes (IRPGs) in PAAD patients; LASSO analysis and the random forest algorithm (RF) were conducted to further determine prognostic-associated gene sets or genes [80], and multivariate analysis was used to assess independent prognostic factors. The performance of the risk model was evaluated using receiver operating characteristic (ROC) curves. The coefficient was obtained from multivariable regression analysis. The risk score was calculated as follows: $\mathrm{risk}\;\mathrm{score}=\sum_{i=1}^{n}\mathrm{expression}\mathrm{of}\;\mathrm{GENE}n*coeffientGENEn$

### 4.10. Chemotherapy Drug Sensitivity Analysis

The “pRRophetic” package based on the GDSC (https://www.cancerrxgene.org, accessed on 15 March 2023) was used to estimate chemotherapy drug sensitivity in PAAD patients [81].

### 4.11. Estimation of Immune Cell Infiltration

The quantification of immune cell infiltration in the PAAD samples was performed using the single-sample GSEA (ssGSEA) algorithm. The identification of marker genes for each infiltrating immune cell was conducted by leveraging validated gene sets from previous research studies [43]. Subsequently, the enrichment scores were examined to determine the levels of immune cell infiltration in patients diagnosed with PAAD. The estimation of immune scores was carried out utilizing the “estimate” algorithm, which uses gene expression data to infer the levels of infiltrating immune and stromal cells in tumor tissues, while also considering tissue purity [82].

### 4.12. ScRNA-Seq Data Processing

We downloaded the pancreatic cancer single-cell sequencing dataset GSE212966 from the GEO database, extracted data from six patients with primary pancreatic cancer in the dataset, and analyzed them using the R package “Seurat”. Low-quality cells were identified and filtered using the following criteria: (1) nFeatures < 200 and nFeature > 2500; (2) mitochondrial gene ratio > 10%. Data were log-normalized using the function “LogNormalize” with the default parameters. To assess cell heterogeneity, we considered the 2000 genes with the highest variability using the “FindVariableFeatures” function with the “vst” method. To analyze the data effectively, a linear transformation was initially applied. Subsequently, dimensionality reduction was carried out in the uMAP space via the utilization of the leading 20 principal components obtained from the PCA. For data clustering purposes, the Seurat graph-based approach was employed, employing a resolution value of 0.8. A total of 30629 high-quality cells were used for analysis, and we identified 9 cell clusters: B cell, Endothelial cells, Neutrophils, T cells, Chondrocytes, Monocyte, NK_cell, Tissue stem cells, and Epithelial cells.

### 4.13. Statistical Analyses

Survival analysis was conducted using the “survminer” and “survival” packages; survival curves were analyzed using the log-rank test. Correlation analysis was conducted using Spearman’s correlation coefficient. Statistical analysis was performed using R studio (R version 4.1.1). The R package “maftools” was used to convert somatic mutation data files into mutation annotation format (MAF) files, considering the mutation frequency of each gene. All experiments were conducted three times independently. The data (mean ± SD) were evaluated using GraphPad Prism version 8.0.2. For data with a normal distribution, we used the t-test (for comparisons between two groups) and ANOVA (for comparisons between three or more groups). If the data did not follow a normal distribution, the Wilcoxon rank-sum test was employed for a two-group comparison and Kruskal–Wallis test for comparisons among multiple groups. Statistical significance was defined as *p* < 0.05.

## 5. Conclusions

In summary, we determined three immune-related subtypes of PAAD according to the variation of immunologic signature gene sets in cancer and adjacent normal tissues and found that genes from tumor gene sets were intimately linked with PC and the TGF-β signaling pathway. The present study underlined the prognostic role of immune-related gene sets in PAAD and constructed an IRPM based on immune-related prognostic genes that showed an excellent predictive efficacy in OS for patients with PAAD, which was successfully validated by an external cohort. We comprehensively analyzed the somatic mutational landscapes and immune landscapes of PC patients with different IRPM risk scores, and we found that our IRPM scores were able to accurately stratify patients according to their immune microenvironment and predict immunotherapy responses, which has important implications for clinicians to develop more targeted strategies.

## Figures and Tables

**Figure 1 ijms-25-00142-f001:**
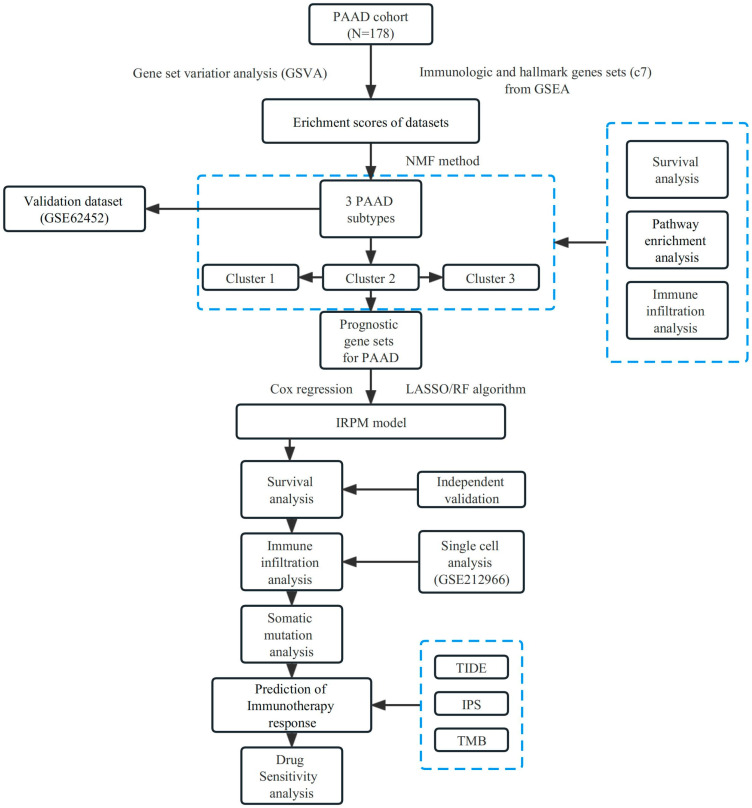
Flow chart of our study design.

**Figure 2 ijms-25-00142-f002:**
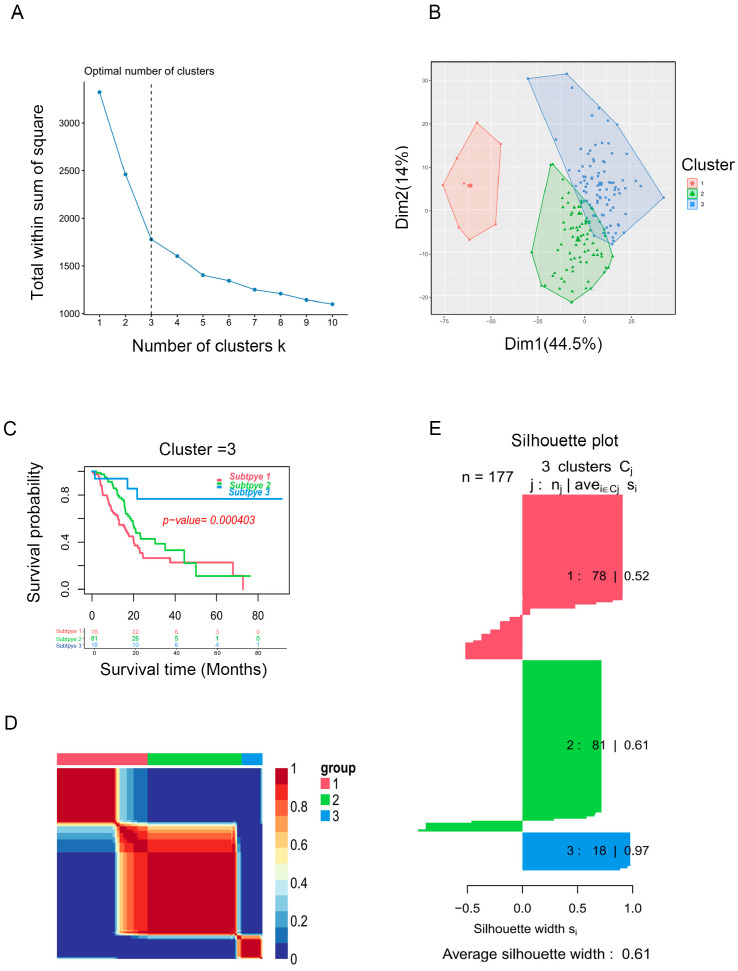
Identification of PAAD subtypes. (**A**) K = 3 was the optimal suggested value of number of clusters. (**B**) Visualization of the cluster results for PAAD samples. (**C**) Survival analysis for PAAD patients among clusters. (**D**) Consensus map of NMF clustering. (**E**) Silhouette plot for the PAAD clusters.

**Figure 3 ijms-25-00142-f003:**
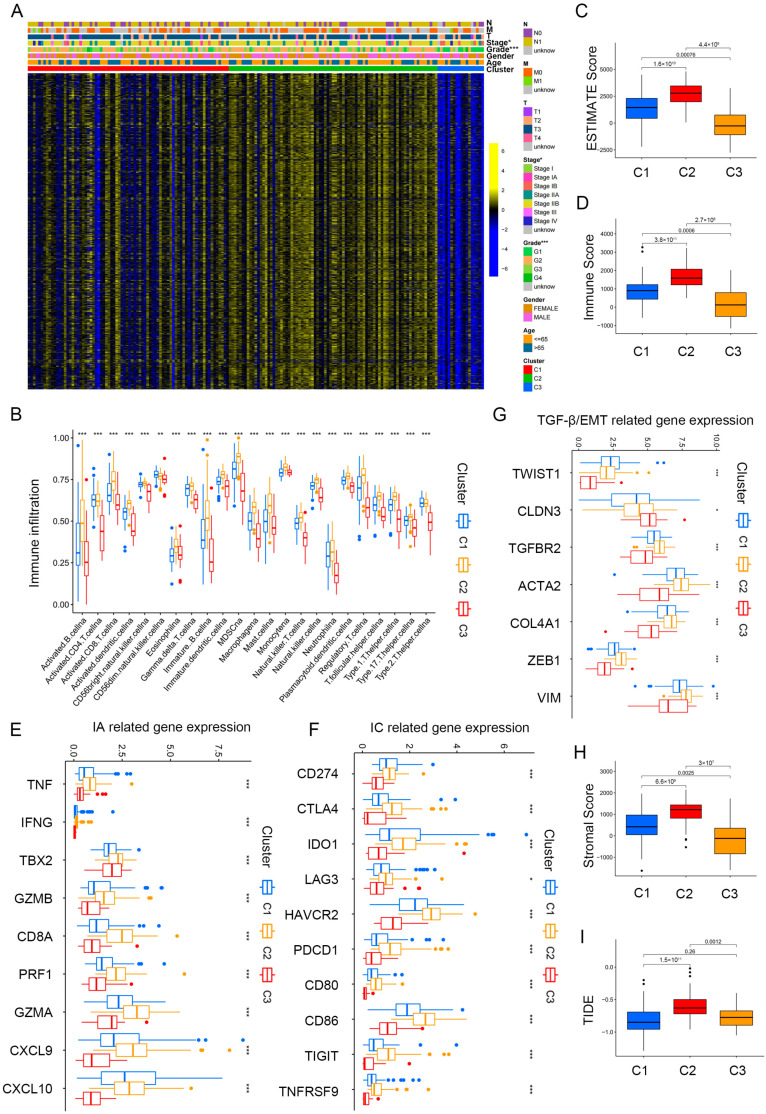
Immune landscape in PAAD patients among three clusters. (**A**) Heatmap demonstrating the clinicopathologic features among three clusters. (**B**) The infiltration levels of 22 immune cells in the three clusters. (**C**,**D**) The ESTIMATE score (**C**) and Immunoscore (**D**) in clusters 1, 2, and 3. (**E**–**G**) The expression levels of immune activation (**E**), immune checkpoint related genes (**F**), and TGF-β/EMT signaling pathway-related genes (**G**) in the three clusters. (**H**,**I**) Differences in stromal score (**H**) and TIDE scores (**I**) in clusters 1, 2, and 3. * *p* < 0.05, ** *p* < 0.01, and *** *p* < 0.001.

**Figure 4 ijms-25-00142-f004:**
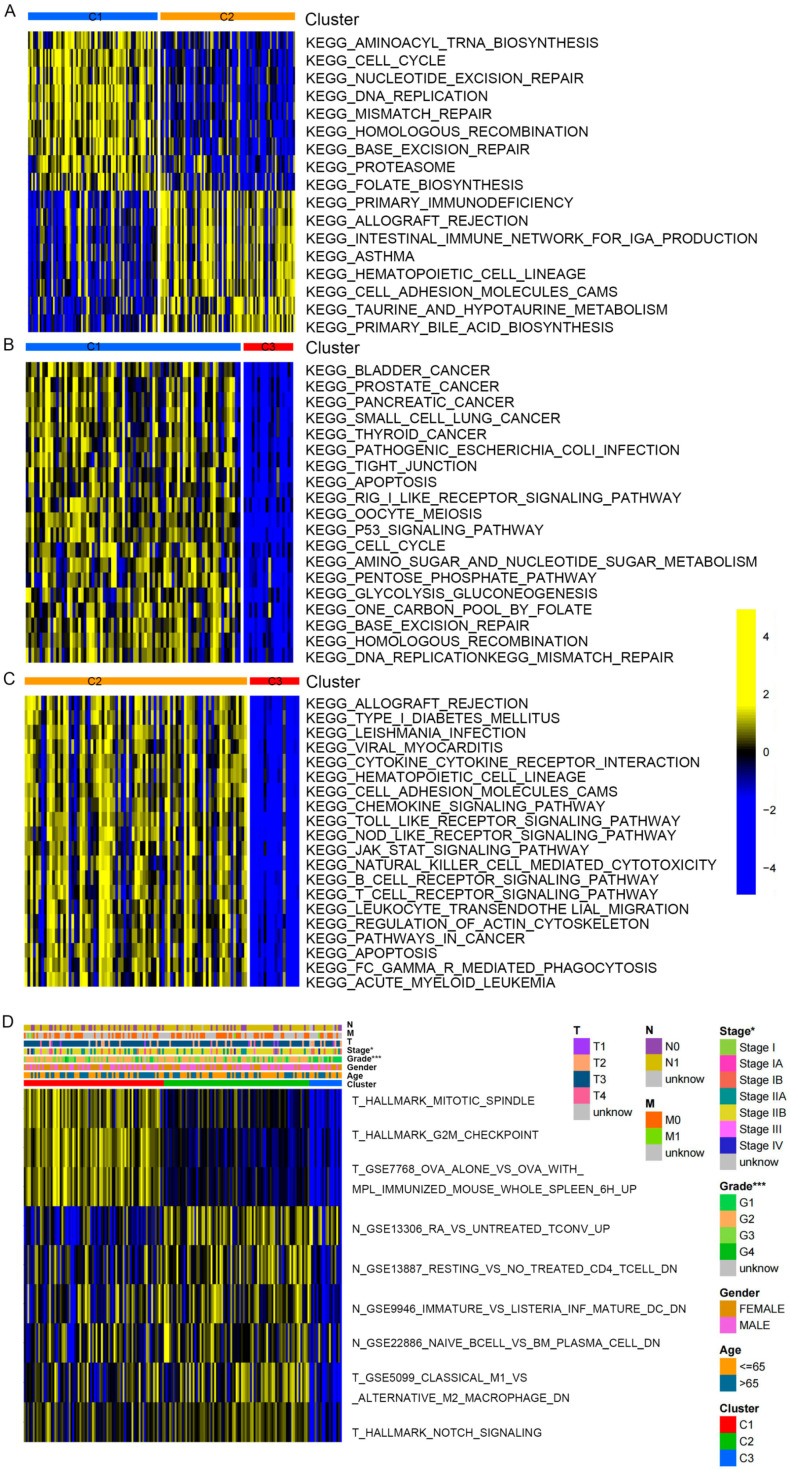
Enrichment score difference of distinct PAAD clusters. (**A**–**C**) KEGG enrichment analysis showing the activation states of signaling pathway in distinct PAAD clusters. (**A**) Cluster 1 vs. cluster 2; (**B**) cluster 1 vs. cluster 3; (**C**) cluster 2 vs. cluster 3. (**D**) Heatmap shows the differential enrichment score of gene sets among 3 clusters. * *p* < 0.05, and *** *p* < 0.001.

**Figure 5 ijms-25-00142-f005:**
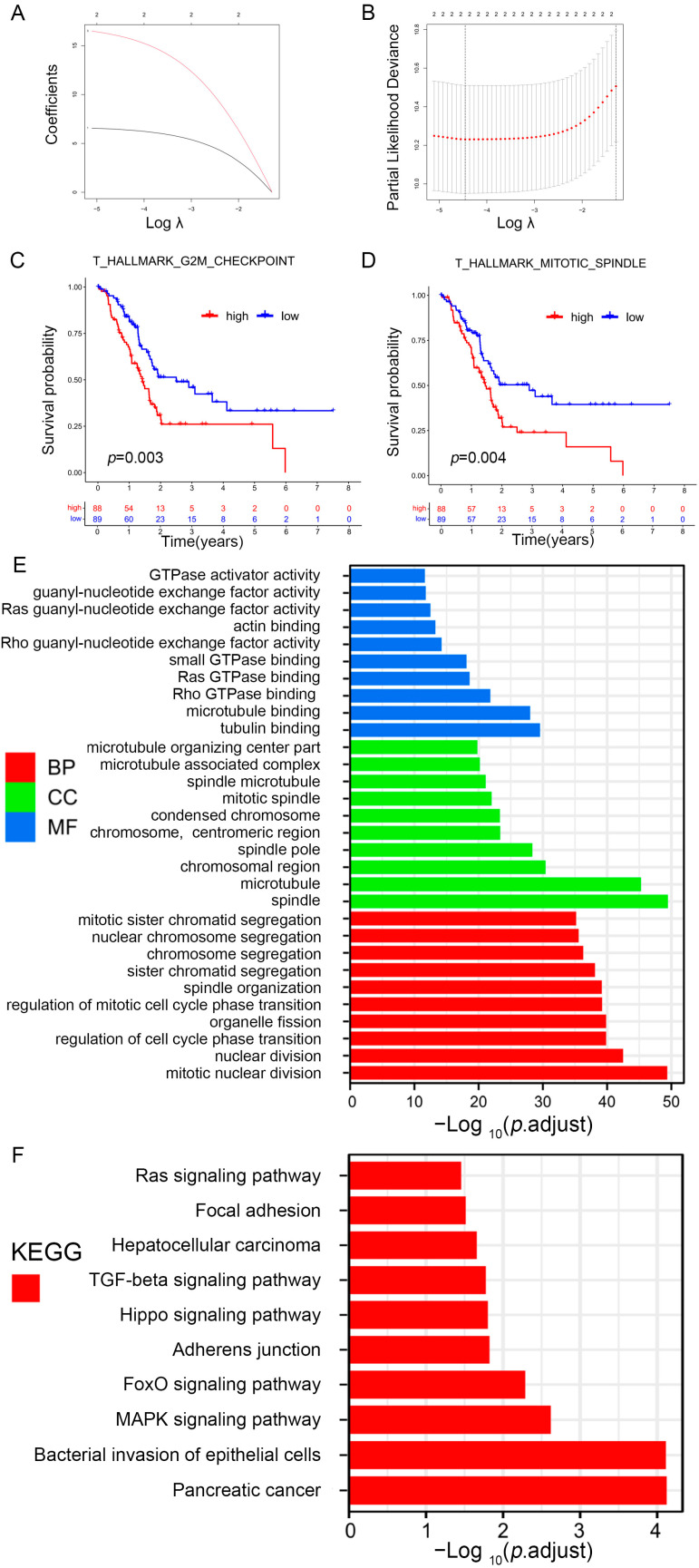
Identification of prognosis-related gene sets in PAAD. (**A**) LASSO coefficient (y-axis) of gene sets and the optimal penalization coefficient (λ) via 10-fold cross-validation based on partial likelihood deviance. (**B**) The dotted vertical lines represent the optimal values of λ. The top x-axis has the numbers of gene sets, whereas the lower x-axis revealed the log (λ). (**C**,**D**) Survival analysis for PAAD patients based on high or low enrichment scores. (**E**,**F**) Visualization for the results of GO (**E**) and KEGG pathway analyses (**F**).

**Figure 6 ijms-25-00142-f006:**
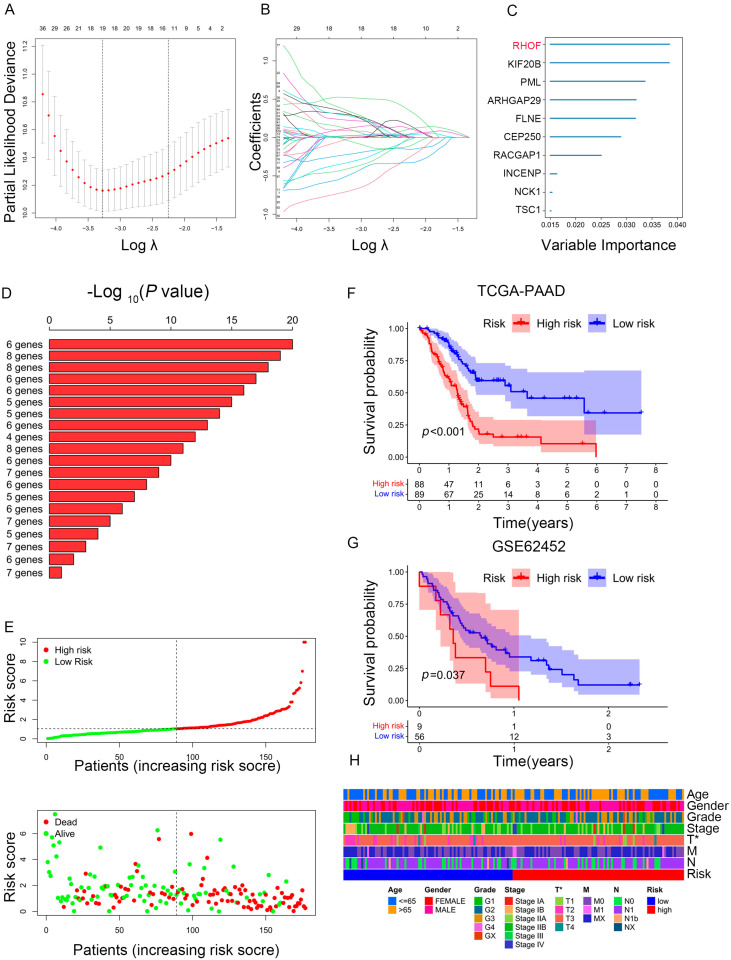
Construction of prognostic model based on prognosis-related genes in PAAD. (**A**,**B**) LASSO coefficient of prognosis-related genes in PAAD. (**C**) Random survival forest analysis screening of 10 genes. (**D**) The top 20 signatures were arranged by the *p* value of KM following Kaplan-Meier analysis of 2^10^-1 = 1023 combinations. As a result of the signature’s relatively high log10 *p*-value and limited number of genes, six genes were filtered out. (**E**) The risk scores plot and overall survival status of prognostic model. (**F**,**G**) Survival analysis for PAAD patients based on high- or low-risk scores in TCGA cohort (**F**) and external validation dataset (GSE62452) (**G**). (**H**) Heatmap depicting the clinicopathologic features between high- and low-risk groups. * *p* < 0.05.

**Figure 7 ijms-25-00142-f007:**
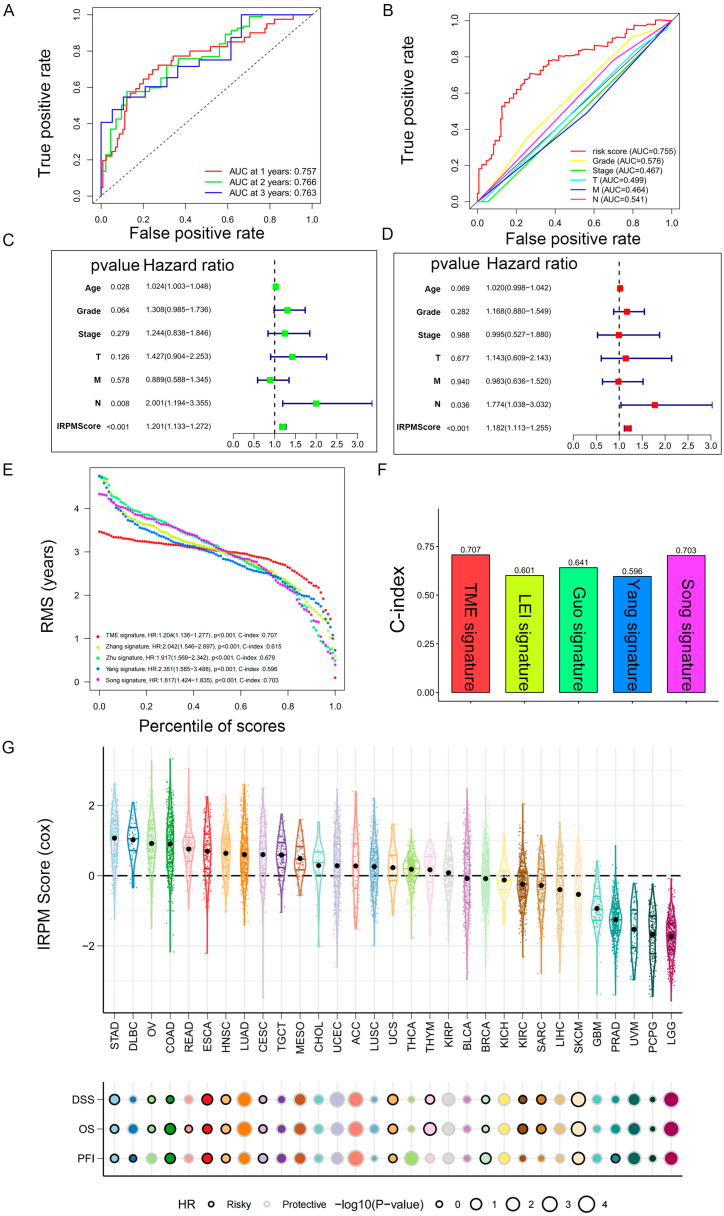
Prediction performance and independence evaluation of IRPM model. (**A**) ROC curve for 1-year, 2-year, and 3-year overall survival. (**B**) ROC curve of IRPM model and other clinicopathological features. (**C**,**D**) Validation of the independence of the risk model in the overall survival through univariate (**C**) and multivariate (**D**) Cox regression analyses. (**E**,**F**) RMS curves (**E**) and C-index (**F**) of IRPM and other previously developed PAAD signatures. (**G**) Prognostic performance of the IRPM model in pan-cancer in TCGA cohort.

**Figure 8 ijms-25-00142-f008:**
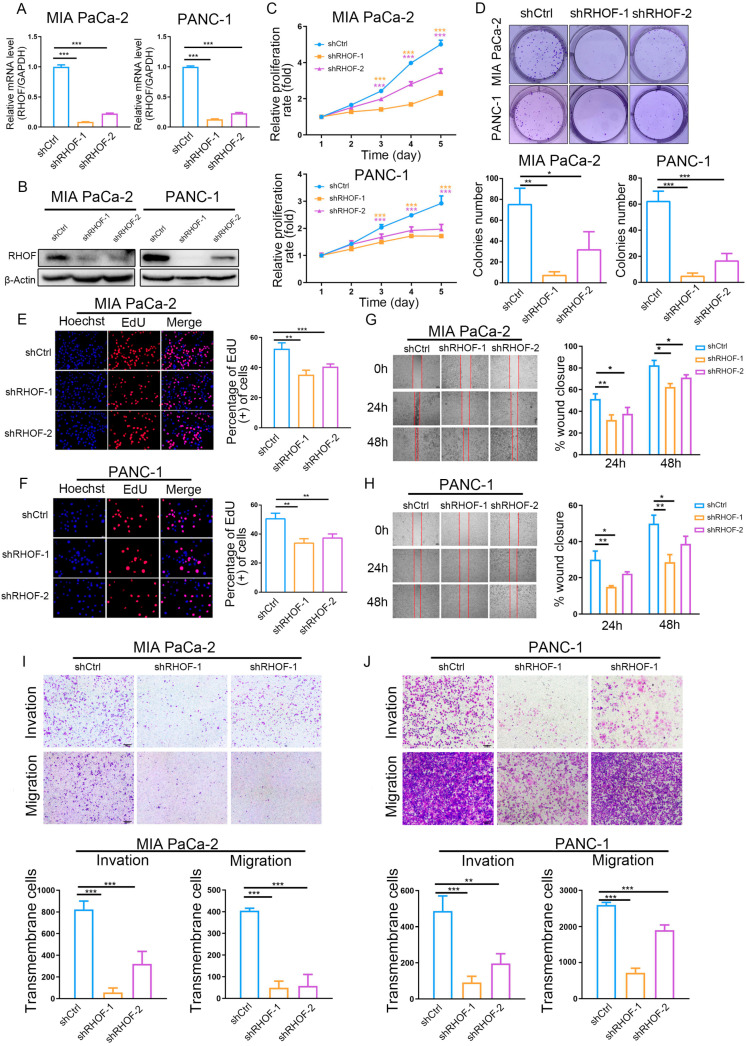
Knockdown of RHOF inhibited the proliferation and migration of PC cells in vitro. (**A**) Relative RHOF expression in PANC-1 and MIAPaCa-2 cells transfected with two independent shRNAs targeting MIR181A2HG by qPCR. (**B**) Western blotting analysis of RHOF protein level in PANC-1 and MIAPaCa-2 cells infected with shRNAs. (**C**) PANC-1 and MIAPaCa-2 cell proliferation after knockdown of RHOF by MTT assay. (**D**–**F**) Representative results of the colony formation (scale bar: 100 μm) and EdU assays (scale bar: 20 μm) in PANC-1 and MIAPaCa-2 cells after RHOF-sh1 or RHOF-sh2 transfection. (**G**–**J**) Representative images of PC cell migration ability as shown by wound-healing assays (**G**,**H**, scale bar: 100 μm) and migration assay (**I**,**J**, scale bar: 200 μm). * *p* < 0.05; ** *p* < 0.01; *** *p* < 0.001.

**Figure 9 ijms-25-00142-f009:**
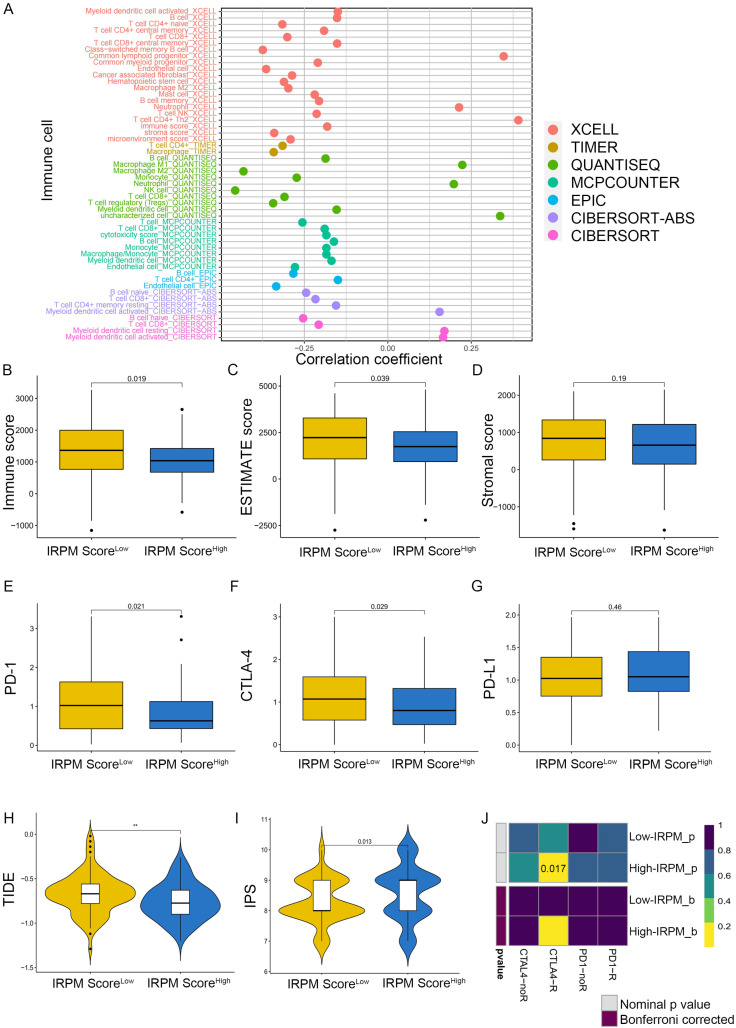
Immune landscape in PAAD patients with different IRPM scores. (**A**) The correlation analysis between IRPM scores and immune cell infiltration in PAAD. (**B**–**D**) Immune score, estimate score, and stromal score in high- and low-IRPM groups. (**E**–**G**) The expression of PD-1 (**E**), CTLA-4 (**F**), and CD274 (**G**) in high- and low-IRPM groups. (**H**,**I**) Differences in TIDE and IPS scores between low- and high-IRPM groups. (**J**) Response to immunotherapy of high- and low-IRPM groups in TCGA, as assessed by the SubMap module of the GenePattern database.** *p* < 0.01.

**Figure 10 ijms-25-00142-f010:**
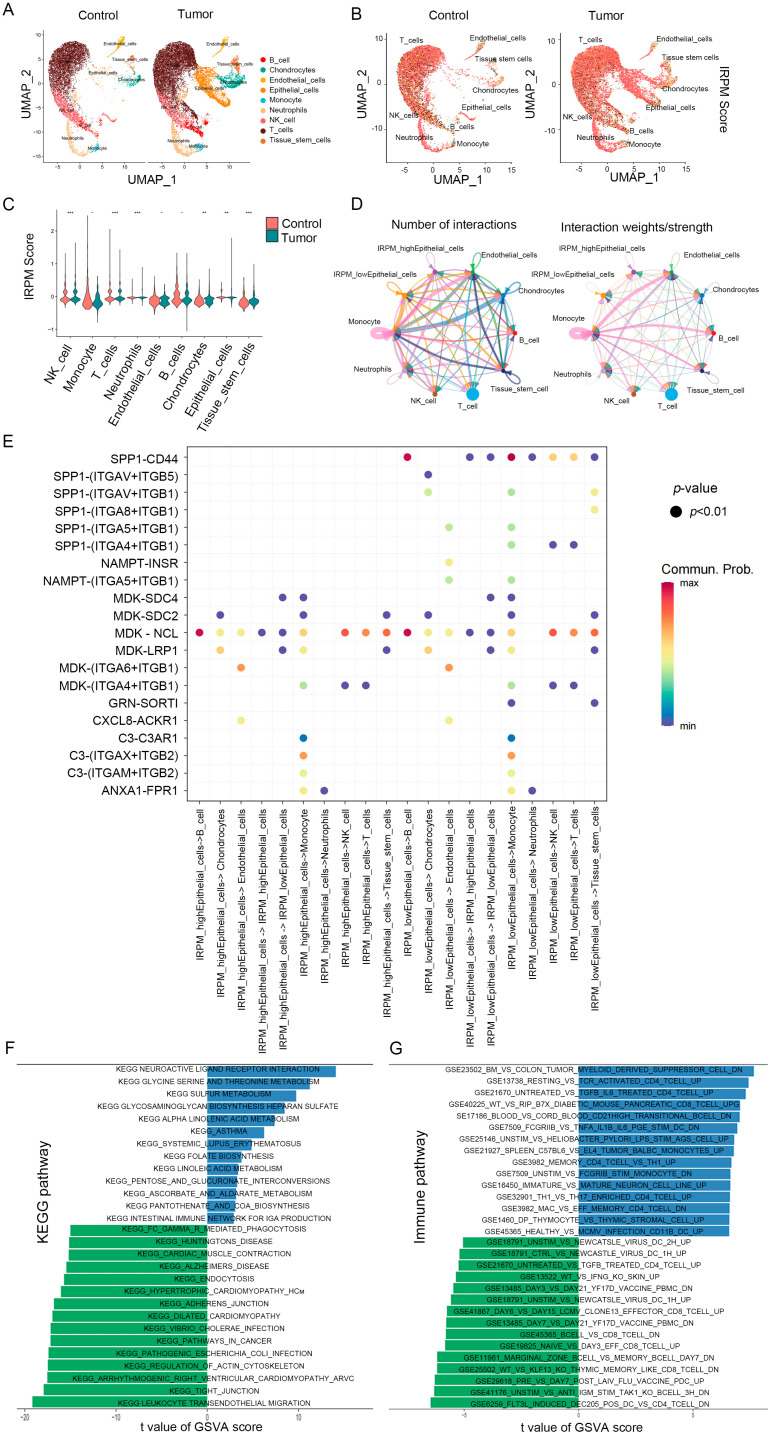
IRPM score heterogeneity and associated immune characteristics in tumor microenvironments. (**A**,**B**) The UMAP visualization shows nine main cell types in a single-cell pancreatic cancer data (GSE2129664), colored by cell type (**A**) and the IRPM scores (**B**). (**C**) Violin plot for the IRPM scores in different cell subtypes. (**D**) Circle plots depicting the interaction numbers and interaction strength between high-IRPM epithelial cells, low-IRPM epithelial cells, and other cell subtypes, respectively. (**E**) Dot plot illustrating substantial ligand-receptor interactions between high-IRPM/low-IRPM epithelial cells and surrounding cells. The CellChat program used colors to represent the communication possibility. The permutation *p*-value is shown by point size. (**F**,**G**) KEGG (**F**) and immune-related (**G**) pathways enrichment analysis of DEGs in high- and low-IRPM score groups. ** *p* < 0.01, and *** *p* < 0.001.

**Figure 11 ijms-25-00142-f011:**
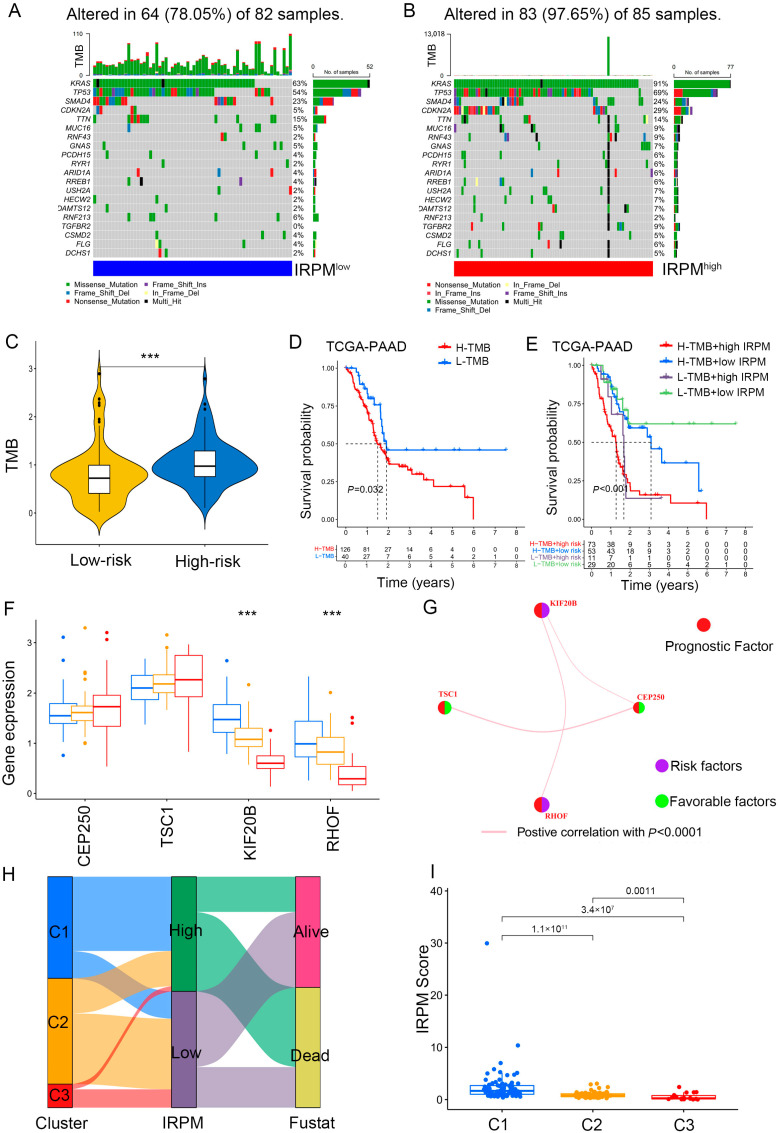
Comparison of somatic mutation landscape in PAAD patients with different IRPM risk scores. (**A**,**B**) The mutation landscape between low- and high-IRPM PAAD samples. (**C**) Differences in TMB between low- and high-IRPM groups. (**D**) Survival analysis for high-TMB and low-TMB groups of the PAAD patients. (**E**) Survival analysis for PAAD patients stratified by both IRPM scores and TMB. (**F**) The expression of 4 immune-related prognostic genes in three gene cluster. (**G**) The interaction between 4 immune-related prognostic genes in PAAD; the size of the circle represents the effect of each gene on the prognosis; purple dots in the circle represent risk factors of prognosis, and green dots represent protective factors of prognosis; the lines linking genes show their interactions; pink represents positive correlation. (**H**) The alluvial diagram of three clusters in groups with different IRPM risk scores and survival status. (**I**) The IRPM scores varied among three clusters. *** *p* < 0.001.

**Figure 12 ijms-25-00142-f012:**
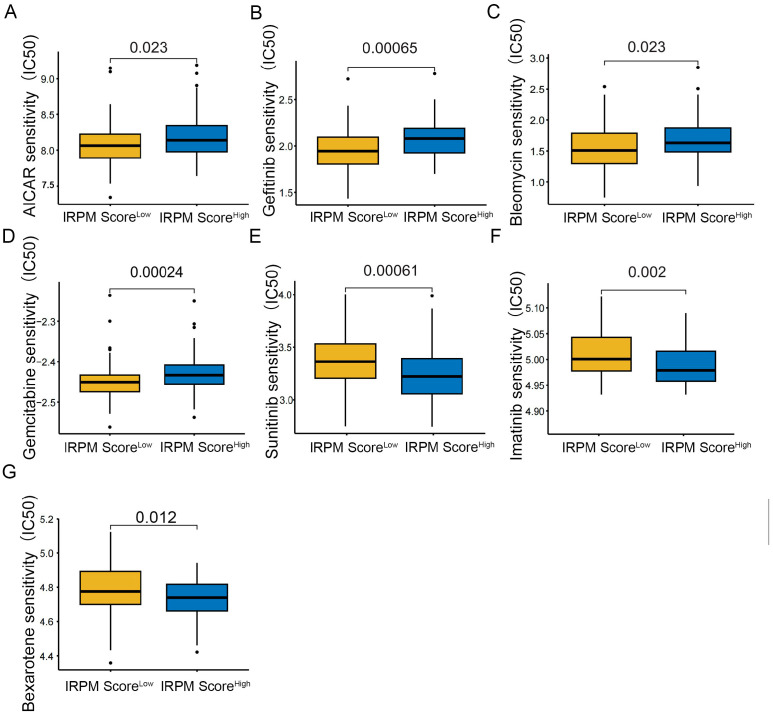
Compared antitumor drug sensitivity in PAAD patients with low- and high-IRPM score groups. (**A**) AICAR, (**B**) Gefitinib, (**C**) Bleomycin, (**D**) Gemcitabine, (**E**) Sunitinib, (**F**) Imatinib, (**G**) and Bexarotene.

## Data Availability

The original contributions in the study are included in the articles/Appendix A. For further inquiry, please contact the corresponding author.

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
