# Peer review of "An Integrated Multi-Omics Analysis Identifying Immune Subtypes of Pancreatic Cancer"

_ijms, 2023, doi:10.3390/ijms25010142_

Round 1
Reviewer 1 Report
Comments and Suggestions for Authors
This study delves into the exploration of putative distinct immune-subtypes within pancreatic cancer and identifies three clusters based from C7 immunological gene-set. The authors claim to have developed an immune-related prognostic risk model (IRPM) with a strong prognostic potential and capable to predict immune-checkpoint blockade response. Notably, the authors identified gene RHOF as a key component of the prognostic model. Moreover RHOF was found to significantly impact pancreatic cancer cells growth and migration.
However, the data are poorly presented, and most of the claims are not supported by the results.
Here are important concerns about your data quality, bioinformatic analysis, methodology and functional analysis:
1. Please review you introduction because I think that it’s not properly on point, moreover your results are not fulfilling your hypothesis.
2. The methods are poorly described. Please include more details about all the methodology that has been used and provide all the missing information about the single-cell analysis.
3. You claimed that you used GSE42568 dataset as external validation cohort. However, these data is from breast cancer samples. I hope that it is an editing error otherwise this represent the major flaw of the study.
4. All the Statistical Analysis are poor and not accurate. I suggest to use the appropriate statistical tests both for normal and non-normal distribution.
5. I think that the C7 immunologic gene set is not appropriate for stratifying PAAD patient based on their immunologic profile. This signature has been obtained from perturbation studies. How can you expect to obtain immunological infiltrate scores from this dataset? Why don’t you haven’t used Immune deconvolution algorithm?
6. I have serious concerns about the outliers, the distribution of your data on Fig.1 and about statistical analysis you conducted
7. I think that RHOF ‘s functional experiments are not on point. You claim that RHOF (along with CEP250, TSC1, and KIF20B) is an immune-related gene associated with prognosis but validation and functional experiments about its role in modulating immunological activity are missing.
8. Please add the cumulative risk tables under all the KM curves. It is not clear the cohorts included in the analysis of fig 11D and E. I have strong concerns about the statistical analysis showed in the entire mauscript.
Comments on the Quality of English LanguageExtensive editing of English language required
Reviewer 2 Report
Comments and Suggestions for Authors
The authors demonstrated that they found clinically relevant immune subtypes of pancreatic cancer. This paper suffers from a proper sample selection issue in the discovery cohort(TCGA-PAAD). It's well-known that the original data of TCGA-PAAD (n=183) have a contamination issue with non-pancreatic cancer tissue such as normal pancreatic tissue or pancreatitis or neuroendocrine tumor(Nicolle, Remy, et al. "Prognostic biomarkers in pancreatic cancer: avoiding errata when using the TCGA dataset." Cancers 11.1 (2019): 126.). over 30 samples in TCGA-PAAD cohort were suspected with this contamination issue, so landmark paper from TCGA-PAAD working group (Raphael, Benjamin J., et al. "Integrated genomic characterization of pancreatic ductal adenocarcinoma." Cancer cell 32.2 (2017): 185-203.) used only 150 samples for their integrative omics analysis. Subtype 3 in Figure 2-C is strongly suspected the sample subset with contamination issue. The authors should analysis again with pure pancreatic cancer samples.
Comments on the Quality of English LanguageEnglish editing is needed.
Round 2
Reviewer 1 Report
Comments and Suggestions for Authors
The authors improved the manuscript.
Comments on the Quality of English Language
Minor editing of English language required
Author Response
Minor editing of English language required.
Response: We thank the reviewer for the helpful comment. We have polished the language carefully. Moreover, the revised manuscript has been edited by an English language editing company (editage).
Reviewer 2 Report
Comments and Suggestions for Authors
The authors didn't show the analysis result according to review comments. In the author's response letter, they said that "even after the removal of these samples and using solely pure pancreatic cancer samples, the conclusions derived from the new analyses remained consistent with the findings previously outlined in our article." But they didn't show any results for this. If the authors want to keep the original submitted paper, they should prove that there is no difference in analysis results even in non-contaminated samples in TCGA-PAAD (n=150).
Comments on the Quality of English LanguageNone
